# A Biomarker Panel Based upon AFP, Fucosylated Kininogen and PEG-Precipitated IgG Is Highly Accurate for the Early Detection Hepatocellular Carcinoma in Patients with Cirrhosis in Phase II and Phase III Biomarker Evaluation

**DOI:** 10.3390/cancers14235970

**Published:** 2022-12-02

**Authors:** Mengjun Wang, Amit G. Singal, Neehar Parikh, Yuko Kono, Jorge Marrero, Anand Mehta

**Affiliations:** 1Basic Science Building Room 310, Department of Cell and Molecular Pharmacology, Medical University of South Carolina, 173 Ashley Avenue, Charleston, SC 29425, USA; 2Division of Digestive and Liver Diseases, University of Texas Southwestern, 5959 Harry Hines Blvd POB I Suite 420B, Dallas, TX 75201, USA; 3Department of Internal Medicine, University of Michigan, Ann Arbor, MI 48109, USA; 4Division of Gastroenterology, University of California at San Diego, 9500 Gilman Drive, La Jolla, CA 92093, USA; 5Division of Gastroenterology, University of Pennsylvania, 3400 Civic Center Boulevard South Pavilion, 4th Floor, Philadelphia, PA 19104, USA

**Keywords:** biomarker, hepatocellular carcinoma, glycosylation, immunoglobulin, human

## Abstract

**Simple Summary:**

Hepatocellular carcinoma (HCC) is one of the most common malignancies worldwide, and the incidence in the United States (USA) is increasing. Generally curative therapies are only possible for early-stage cancers, thus biomarkers that can be used to detect early-stage cancers are needed. Here, we describe the identification of a new biomarker that when used as a part of an algorithm can be used for the early detection of HCC with high accuracy.

**Abstract:**

We have previously identified alterations in glycosylation on serum proteins from patients with HCC and developed plate-based assays using lectins to detect the change in glycosylation. However, heterophilic antibodies, which increase with non-malignant liver disease, compromised these assays. To address this, we developed a method of polyethylene glycol (PEG) precipitation that removed the contaminating IgG and IgM but allowed for the lectin detection of the relevant glycoprotein. We found that this PEG-precipitated material itself could differentiate between cirrhosis and HCC. In the analysis of three training cohorts and one validation cohort, consisting of 571 patients, PEG-IgG had AUC values that ranged from 0.713 to 0.810. In the validation cohort, which contained samples from patients at a time of 1–6 months prior to HCC detection or 7+ months prior to detection, the AUC of this marker remained consistent (0.813 and 0.846, respectively). When this marker was incorporated into a biomarker algorithm that also consisted of AFP and fucosylated kininogen, the AUROC increased to 0.816–0.883 in the training cohort and was 0.909 in the external validation cohort. Biomarker performance was also examined though the analysis of partial ROC curves, at false positive values less than 10% (90-ROC), ≤20% (80-ROC) or ≤30% (70-ROC), which highlighted the algorithm’s improvement over the individual markers at clinically relevant specificity values.

## 1. Introduction

Hepatocellular carcinoma (HCC) is one of the most common malignancies worldwide and the incidence in the United States (USA) is increasing [1,2,3,4,5,6]. Patients with cirrhosis from any etiology have a high risk of developing HCC, with an annual incidence of 1–3% per year. Given the strong association between early detection and improved survival, patients are recommended to undergo semi-annual surveillance, usually with ultrasound plus serum levels of alpha-fetoprotein (AFP) [7]. The progression of liver disease into liver cancer can also be monitored with imaging modalities such as ultrasound, magnetic resonance imaging (MRI) and computed tomography scan (CT-scan). However, ultrasound plus AFP’s limited sensitivity and specificity has reduced its reliability as a primary screening modality for HCC [8,9], and more sensitive biomarkers for HCC are desired [6]. 

Using a glycoproteomic approach, we previously identified serum glycoproteins that had increased fucosylation in HCC [10], and validated the biomarker potential of many of these in multiple patient sets using simple plate-based assays [11]. However, these plate-based assays were challenging to perform, as pre-malignant liver disease (fibrosis) was associated with increased levels of heterophilic antibodies, which themselves had altered glycosylation that made them reactive to the lectins that we were using [12]. To overcome this technical challenge, we utilized several methods to remove these heterophilic immunoglobulins from serum [13,14]. One method developed utilized polyethylene glycol (PEG) to deplete heterophilic immunoglobulins from serum [14]. Unexpectedly this PEG-precipitated material, referred to as PEG-IgG, had the ability to differentiate between cirrhosis and HCC. It is important to note that IgG only had discriminatory ability after PEG precipitation. Herein, we report results examining biomarker performance in three cross sectional samples of patients with and without HCC from both viral and non-viral etiologies (biomarker Phase 2) and samples collected using a prospective specimen-collection, retrospective-blinded-evaluation (PRoBE) design (biomarker Phase 3) [15]. 

## 2. Materials and Methods

### 2.1. Patient Samples and Ethics Statements 

Four independent serum sample sets were utilized for this study (Table 1). The first was from the University of Texas Southwestern (UTSW) Medical Center (*n* = 61), the second set was from St. Louis University (*n* = 74), the third was from the University of California at San Diego (*n* = 339) and the fourth was from the University of Michigan (*n* = 97). Serum samples were obtained via a study protocol approved by the appropriate Institutional Review Board. In all cases, written informed consent was obtained from each subject. Diagnosis of cirrhosis was based on liver histology or clinical, laboratory and imaging evidence of hepatic decompensation or portal hypertension (15). Each non-HCC patient had a normal ultrasound; if serum AFP were elevated, a CT or MRI showed no liver mass. For HCC patients, the diagnosis of HCC was made per AASLD guidelines [16] using histopathology or characteristic imaging (magnetic resonance imaging [MRI] or computed tomography [CT]) showing a vascular enhancing mass with delayed washout) (5). Early-stage HCC was defined using the Milan Criteria [17]. Demographic and clinical information were obtained, and a blood sample was collected from each subject. A 20-mL blood sample was drawn from each subject, spun, aliquoted, and serum stored at −80 °C until testing. Blood samples for HCC patients were drawn prior to initiation of HCC treatment. The demographic of each cohort was presented in Table 1. 

### 2.2. PEG-Precipitation of Serum

HCC occurs in the background of liver fibrosis (cirrhosis), and liver fibrosis is associated with altered glycosylation of heterophilic antibodies [13,18,19]. This change in glycosylation alters their binding to fucose binding lectins and methods were developed to remove these from serum prior to analysis of fucosylated proteins. Briefly, in the optimized PEG method, 15 µL of serum was mixed with 5 µL PBS and incubated with 20 µL of 40% polyethylene glycol (PEG)-8000 (VWR, Radnor, PA, USA, Cat. 0159-2.5KG) for 30 min at room temperature with shaking, and the samples centrifuged at 14,000× *g* for 30 min at 4 °C. Fucosylated kininogen (as well as many other fucosylated proteins of interest) was contained in the supernatant, while the immunoglobulins, was found in the pellet. The PEG-precipitated material was suspended with PBS and used for IgG analysis. 

### 2.3. Analysis of IgG Levels in PEG-Precipitated Material with Indirect, Direct, Competitive and Sandwhich ELISA

PEG-precipitated IgG was re-suspended with 1.5 mL of PBS by setting in an Eppendorf Mixmate (Eppendorf, Enfield, CT, USA), at 1500 rpm for 30 min. Subsequently, 10 µL of this suspension (equivalent 0.1 µL serum) was mixed with 90 µL of 1× PBS, transferred to a well of an ELISA plate (NUNC-Immuno Module, Thermo Scientific, Waltham, MA, USA, Cat. 469949), at 4 °C overnight. The next day, the plate was washed 3 times with PBST (0.1) and blocked with 1× Carbon-free blocking (Vector Lab, Newark, CA, USA, Cat. SP-5040-125), at 37 °C for one hour. After washing, IgG was detected using an anti-Human IgG antibody (Rockland, Pottstown, PA, USA. Cat. 609-4102. Dilution 1:10,000), incubated at 37 °C for one hour. Bound anti-human IgG was detected using a secondary IRDye-labelled animal specific antibody (LI-Cor, Lincoln, NE, USA, Cat. 926-32211, Dilution 1:10,000), incubated at room temperature for 45 min and signal detected and quantified with an Odyssey DLx Imaging System (LI-Cor, Lincoln, NE, USA). Normal serum was set as internal control (Sigma-Aldrich, Burlington, MA, USA, Cat. H4522).

PEG-IgG was also detected using a commercially available IgG sandwich ELSIA kit (Catalogue #ab100547, abcam, Waltham, MA, USA). Additionally, PEG-precipitated IgG was also examined using IgG subtype kits (Invitrogen, Waltham, MA, USA), to quantify total IgG, IgG1, IgG2, IgG3, IgG4 in serum or in the PEG pellet. All procedures followed the manual of the kit (catalogue numbers BMS2091 (total IgG), BMS2092 (IgG1), BMS2093 (IgG2), BMS2094 (IgG3), MS2095 IgG4). 

PEG-precipitated IgG was also examined using a competitive ELISA system. Briefly, commercially purified human IgG (Sigma-Aldrich, Burlington, MA, USA, Cat. I4506) was labelled with IRDye 800CW (LiCor, Lincoln, NE, USA, Cat. 928-38040). Ninety-six-well plates were coated with the anti-human IgG antibody (500 ng/100 µL PBS) used in the indirect ELISA at 4 °C overnight, and the next day the sample wells were washed and blocked in 1× Carbon-free blocking buffer at 37 °C as above. A mixture of PEG-IgG (equivalent to 0.1 µL serum) and 3 µg of the labelled IgG was added to the plate. The intensity of the spiked and labelled human IgG to out compete the unlabelled serum derived IgG was quantified with the Odyssey DLx Imaging System as above. 

Lastly, PEG-precipitated IgG was also examined by Western blot following PEG-precipitation. In this situation, we simply mixed loading buffer with desired amount of PEG-precipitated IgG and resolved bands using 12% polyacrymide gel to electrophoresis, and stained the total protein Coomassie brilliant blue (SimplyBlue SafeStain, Invitrogen, Waltham, MA, USA, Cat. LC6060), or by immunoblot detection. In all cases, visualizing and quantification were done with the Odyssey DLx Imaging System.

### 2.4. Analysis of Fucosylated Kininogen

Our plate-based assays for the analysis of fucosylated kininogen and other proteins is as described elsewhere [14]. 

### 2.5. Statistical Analysis

Descriptive statistics for patient groups were presented in Table 1. Distributions of all variables were reported as mean values ± SD unless otherwise stated. To evaluate performance of classification of each biomarker, or paneled biomarkers, we constructed ROC curves for evaluations, 95% confidence intervals were provided. 

To facilitate comparing biomarker performance from different cohorts, especially for analysis of external-cross-validations, normalization of biomarker values was applied. In this study, z-score normalization was selected, *vi*’ = $\frac{vi-\bar{A}}{\sigma }$,where $\bar{A}$ and σ, are mean and standard deviation, respectively. 

For algorithm development, a logistic regression model was selected as a main tool to construct algorithms; random Forest and xgboost algorithms were applied to validate Logistic regression modeling. The predictive ability of the algorithm was challenged by Repeated-3-fold Cross-Validation, and External-Cross Validation. To avoid the situation in which some samples were never chosen by randomization in the training set, nor test set, we randomly arranged the original dataset and split the new dataset into 3 parts (Repeated 3-fold Cross-Validation) and used two parts for training and one part for validation. We applied this procedure 200 times to guarantee every sample be in both test set and training set. 

Partial ROC construction: ROC curves above a certain specificity (such as 90%, 80% or 70%) have more practical meaning than an analysis of the full range of specificity values. Thus, we developed ROC curves with only data above set specificities values. In this case, ROC curves above 90% specificity were referred to as 90-ROC. Similar naming was used for ROC at specificity values above 80% (80-ROC) and 70% (70-ROC). We estimated these areas under curve with a spline method, which we refer to as a partial area under the curve (Partial AUC), and we also calculated relative areas under curve in predefined area, such as above specificity 90%, referred to as the relative AUC. 

## 3. Results

### 3.1. Identification of PEG-Precipitated IgG as a Biomarker of HCC

Heterophilic IgG and IgM have previously been identified as contaminants in a plate-based lectin assay [13]. A method that utilized polyethylene glycol-8000 (PEG-8000) for the removal of IgM and IgG was developed. Figure 1A shows the general procedure of this assay. Surprisingly, while this method allowed for the analysis of fucosylated target proteins, the IgG in the PEG-precipitated fraction also had biomarker ability. This is shown in Figure 1B through the analysis of PEG-IgG in samples from the University of Texas, that consisted of 40 cirrhotic samples and 21 HCC samples (Table 1). A direct-ELISA for IgG was used on samples following PEG treatment. As Figure 1B shows, the level of detectable IgG following PEG precipitation was higher in the HCC samples compared to the cirrhotic samples (*p* < 0.01). In this sample set, PEG-IgG had an AUROC of 0.711 (95% confidence interval: 0.568 to 0.854, *p* = 0.0066) in differentiating HCC from cirrhosis (Figure 1C). It is important to note that PEG precipitation was essential for biomarker performance. That is, as shown in Appendix A, when we applied a similar procedure of PEG-IgG on serum without PEG precipitation, there was no discriminatory ability to differentiate the HCC from the cirrhotic group (AUC = 0.505, *p* = 0.952). The AUC of the PEG-IgG material was statistically different than the AUC of total IgG (Delong’s test, *p* < 0.001).

To further examine the nature of this PEG-precipitated material, we examined the IgG subtype of the precipitated IgG. Briefly, we utilized a pool of HCC (22 samples), cirrhotic (22 samples), and commercially purchased “healthy” serum for subtype analysis. These three serum samples were precipitated in 10%, 15% or 20% PEG, and the pellets were quantified for their IgG subtypes using a commercially available human IgG subtyping kit. As Appendix A shows, IgG was higher in patients with HCC as compared to healthy or cirrhotic serum, but only when samples were PEG-precipitated. Similarly, and in contrast to IgG1 or IgG4 levels, IgG2 and IgG3 levels were higher in PEG-treated HCC serum as compared to normal cirrhotic sera (compare Appendix A to Appendix A). It is noted that overall biomarker performance was higher when examining total PEG-precipitated IgG as opposed to specific IgG subtypes (Appendix A). In addition, we used several different commercially available anti-human IgG antibodies and obtained similar results (Appendix A). And lastly, we used a sandwich ELISA total IgG quantification kit to examine the entire University of Texas group (AUC = 0.616, *p* = 0.1409; data not shown [20]) and while it had similar results to our direct ELISA (AUC = 0.711, *p* = 0.007) the direct detection of the PEG-IgG had better performance and was utilized going forward.

### 3.2. Validation of PEG-Precipitated IgG as a Biomarker of HCC at Time of Cancer Detection

To further examine the biomarker performance of the PEG-IgG we examined it further two independent patient sample sets. The first sample set (UCSD, Table 1) consisted of 339 total patients: 107 with HCC in the background of cirrhosis, 184 cirrhosis without HCC, and 48 patients treated for HCC (21 by transplant and 27 by resection or ablation). The second sample set consisted of 40 cirrhosis without HCC and 34 cirrhosis with early-stage HCC (St. Louis University (SLU)). As Figure 2A shows, increased levels of the PEG-IgG marker were observed in the UCSD HCC samples as compared to those with non-HCC cirrhosis. The mean level of PEG-IgG was 1.62, SD = 0.21 in HCC samples and mean = 1.43, SD = 0.21 in cirrhosis samples (*p* < 0.0001, *t*-test). Results were similar to that observed in the initial discovery cohort (University of Texas), mean = 1.52 (SD = 0.37 in HCC samples, Mean = 1.22, SD = 0.34, in cirrhosis patients) (*p* = 0.003). 

In the SLU sample set (Figure 2C), which comprised early-stage HCC samples, the mean level of PEG-IgG was 1.84, SD = 0.37 in HCC samples and 1.35 (SD = 0.37) in cirrhosis samples indicating that this marker was associated with early-stage HCC (*p* < 0.0001). 

The biomarker potential of PEG-IgG was analyzed through AUROC analysis (Figure 2B,D). When comparing cirrhosis patients with and without HCC, the AUROC of this marker in the UCSD sample set was 0.734 (95%CI: 0.675–0.792) and 0.810 (95%CI: 0.712–0.909) in SLU sample set.

Interestingly, when the PEG-IgG marker was examined in patients following HCC treatment (*n* = 48), several interesting observations were noticed. First, the level of PEG-IgG, similar to what is observed with AFP and another potential biomarker, fucosylated kininogen, was reduced in patients who received a liver transplant, as compared to those with HCC (Appendix A). However, in contrast to AFP and fucosylated kininogen, the level of PEG-IgG was not reduced in patients who had undergone a resection (Appendix A), suggesting that this biomarker is not coming from the tumor itself (as are the other biomarkers) but may be associated with liver stromal factors. 

### 3.3. Potential Prognostic Value of PEG-Precipitated IgG

The samples in the previous case–control sample sets were all taken at the time of HCC detection (or close to). To determine how well this marker worked before HCC was detected, we used a prospective specimen-collection, retrospective-blinded-evaluation (PRoBE) design to validate the performance in a study consisting of 37 patients who developed HCC in the background of cirrhosis and 60 cirrhosis patients without HCC. These patients were a subset of a larger cohort that included 442 patients with Child A or B cirrhosis, without HCC at baseline, who were enrolled into a prospective HCC surveillance program and followed every 6 months until time of HCC diagnosis, liver transplantation, or death [21]. Samples were examined at the time of 1–6 months prior to HCC detection (mean of 3.2 months) or at a time of 7+ months prior (mean of 12.1 months). 

Figure 3A shows the level of the PEG-IgG marker in those at a time of 1–6 months prior to HCC detection and Figure 3C shows the level at a time point of 7+ months prior to HCC detection. As these figure shows, at both time points, elevations of PEG-IgG are observed in the HCC samples as compared to those patients with cirrhosis (*p* < 0.001). As Figure 3B shows, an AUC of 0.813 (95%CI: 0.726–0.900) was observed at the 1–6-month time point. The AUC was 0.846, (95%CI: 0.763–0.928) at the 7+ month time point (Figure 3D). This AUC was not statistically different than observed at the earlier time point (*p* = 0.6078, bootstrap test). 

In an effort to fully examine the biomarker performance, we performed AUC analysis of these markers at fixed false positive values of 10%, 20% and 30%. These partial ROCS are referred here as 90-ROC, 80-ROC and 70-ROC. Appendix A show the total AUC, the 90-ROC, 80-ROC and 70-ROC for PEG-IgG, AFP and fucosylated kininogen in this validation cohort. As Appendix A show, all three of these markers have similar overall ROC, 90-ROC, 80-ROC and 70-ROC values. 

### 3.4. Inclusion of PEG-IgG into a Biomarker Algorithm

In an effort to improve the biomarker performance of the PEG-IgG, this marker was incorporated into biomarker algorithm comprising AFP and fucosylated kininogen as we have done previously [14,22,23,24,25]. The general flow of model development is shown in Appendix A but briefly, the data from three “discovery” cohorts were pooled and used to develop a model for classification of HCC. The model, as shown in Appendix A, was composed of PEG-IgG, AFP, and fucosylated kininogen. The model was tested through internal cross validation (3-fold CV, data not shown [20]) of the entire cohort and through individual analysis of the specific cohorts (Appendix A). The same model was subsequently applied on the data from the University of Michigan at the 1–6 month and 7+ month time points (external validation, Figure 4). 

Appendix A highlight the AUROC values for the three individual cross-sectional cohorts. In the cohort from UCSD, the algorithm resulted in an AUROC of 0.838, which was greater than any marker alone (AFP = 0.727). Similar increases in AUROC values were observed in the SLU and UT cohorts. Appendix A also show the partial ROCs (90-ROC, 80-ROC and 70-ROC) highlighting the increased sensitivity at high specificity of the algorithm in each of these cohorts at the time of HCC diagnosis. Importantly, the partial ROCS highlighted the greatest increase in biomarker of the algorithm, which improved the 90-ROC to 0.319 for the algorithm in the UCSD cohort from 0.278 observed with AFP alone and increased the sensitivity to 58% at 90% specificity as compared to 39% sensitivity for AFP alone. 

Figure 4A,B presents the external validation of the algorithm in longitudinal samples either 1–6 months before HCC diagnosis or 7+ months prior to HCC diagnosis. Similar to the cross-sectional data, the AUROC values were higher in both time points than any individual marker alone (See Appendix A) with an AUROC of 0.909 at a time of 1–6 months prior to HCC detection and 0.819 at a time of 7+ months prior to HCC detection. However, the true benefit of the algorithm is best observed through an examination of partial ROCs, as shown in Figure 4C–F. Here, the 90-ROC data are shown. As this figure shows, the algorithm improved performance over the individual markers at specificities 90% or greater, with partial ROC values increasing by close to 60% as compared to the AFP alone (comparing Figure 4C to Figure 4D–F). Similar results were obtained at lower specificity values as well (Appendix A). 

In this longitudinal study, 76% HCC patients, were early-stage as defined by Milan Criteria with a mean tumor diameter of 2.3 cm (range 0.5–4.4 cm). The biomarker algorithm had an AUROC of 0.851 with a 55% sensitivity at 90% specificity at all time point in these patients with only early cancers, which was significantly higher than either marker alone (*p* < 0.001) (Appendix A). As before, ROC-90, ROC-80 and ROC-70 values all showed improved performance of the algorithm as compared to individual biomarkers (compare Figure 5 to Appendix A). 

## 4. Discussion

Alterations in immunoglobulins have been associated with many diseases. Indeed, we and others, have previously identified changes in the level and in the glycosylation on IgG in patients with cirrhosis [26]. These increases in both protein level and the alterations in glycosylation have proven to be confounders in plate-based assays for the analysis of fucosylated proteins in HCC. To that end, we developed a method to remove heterophilic IgG and IgM from serum samples prior to analysis of fucosylated proteins. This method involves the simple addition of PEG-8000 (20% final concentration) to serum and incubation of the mixture on a shaker for 30 min prior to centrifugation. It is noted that this method was optimized in previous work to maximize the removal of heterophilic antibodies with minimal impact upon the fucosylated glycoproteins of interest. The molecular basis for the PEG-mediated precipitation of proteins such as the immunoglobulins [27,28,29] is not fully understood, but may be related to the size and hydrophobicity of the protein [27]. However, there is uncertainty regarding the manner in which PEG precipitates many proteins, including IgG [28].

Here, very surprisingly, we have shown that the PEG-precipitated IgG itself had discriminatory ability in multiple independent cohorts. Importantly, this was true only of PEG-precipitated IgG and not of total non-precipitated IgG (Appendix A). Initial assays were performed using a simple direct ELISA format with PEG-precipitated material deposited onto a 96 plate, followed by IgG detection using an anti-human IgG antibody. These results were subsequently replicated using multiple commercially available anti-human IgG antibodies as well as commercially available human IgG assay kits (Appendix A).

Efforts to further examine this material by sub-type (Appendix A) indicated that greatest discriminatory ability was observed when examining either total IgG or IgG3. The specific nature of this immunoglobulin and the role of the IgG3 subtype in cancer development are under further investigation. It is also noted that while the PEG-precipitated material contained IgM, the IgM levels had no discriminatory ability (data not shown [20]).

It is noted that although PEG-precipitated IgG had discriminatory ability, it was not statistically different than that observed with AFP and other markers. This is not surprising given the heterogeneity of HCC. However, when part of a diagnostic algorithm, it resulted in a substantial improvement over any individual marker. The algorithm that was developed consisted of AFP, PEG-IgG, and fucosylated kininogen (Appendix A). This algorithm was created using a pool of samples that comprised three cohorts (the University of Texas, Saint Louis University, and the University of California San Diego). Subsequently, this algorithm was tested on each individual cohort through internal examination of each individual cohort (Appendix A) and through external validation of a longitudinal cohort of samples containing samples collected at a time point of 1–6 months prior to HCC detection or from samples 7+ months prior to HCC detection (Appendix A). In all cases, the algorithm comprising these three factors increased the discriminatory ability over the biomarkers alone and, importantly, at clinically relevant specificity values.

When compared to our previously identified biomarker algorithms [22,24], the new PEG-IgG-based algorithm had greater AUC values in these patient cohorts both through external validation cohorts and through apparent validation (Appendix A). These results highlight the improved biomarker potential of the PEG-IgG-based algorithms and also imply that further refinement of those other algorithms may be required to improve robustness. One thing of value is that the method for PEG precipitation of IgG is very simple and requires little technical expertise and is a pre-request for the analysis of fucosylated glycoproteins such as fucosylated kininogen. In addition, the subsequent analysis of the PEG-IgG is very simple and follows well established direct ELISA methods.

It is also noted that the PEG-IgG-based algorithm was also greater than that observed with the GALAD algorithm in the same UM patient cohort. That is, GALAD had an AUROC of 0.77 in the same UM cohort, which was exceeded by the PEG-IgG which had an AUROC of 0.91 in those at 1–6 months prior to HCC detection and 0.87 in those with early-stage HCC at any time point (GALAD = 0.79) [30].

Improved algorithm performance was best observed through an examination of partial AUROC, which we refer to as either ROC-90 (ROC at only ≥90% specificity), ROC-80 (ROC at only ≥80% specificity or ROC 70 (ROC at only ≥70% specificity). As Figure 5 shows, the AUROC for the algorithm in the ROC-90 plot increased to 0.484, which was substantially higher than AFP alone (0.303), kininogen alone (0.244) or the PEG-IgG alone (0.225). Similar results were observed in the ROC-80 and ROC-70 plots. Importantly, what these data highlight is that overall sensitivity is increased as compared to AFP, at high specificity values that are clinically relevant, with 72% sensitivity at 90% specificity achieved with the algorithm (as compared to 45% sensitivity at 90% specificity for AFP alone). It is noted that we propose the use of the ROC-90 as the new standard by which biomarkers should be compared, as it provides more clinically relevant data to compare biomarkers.

## 5. Conclusions

In conclusion, we describe the discovery that the levels of PEG-precipitated IgG are higher in the serum of patients with HCC in the background of cirrhosis than in the serum of patients with cirrhosis alone. This assay can be performed using a simple direct ELISA approach, or even a competitive ELISA format, using only a few microliters of serum. Importantly, this marker, when combined with AFP and fucosylated kininogen, which we have described previously, resulted in a simple algorithm that dramatically increased the discriminatory ability as compared to either marker alone or other biomarker algorithms previously identified. Future work will need to examine this biomarker in combination with imaging methods such as ultrasound and Abbreviated MRI to determine the true clinical benefit of this algorithm.

## Figures and Tables

**Figure 1 cancers-14-05970-f001:**
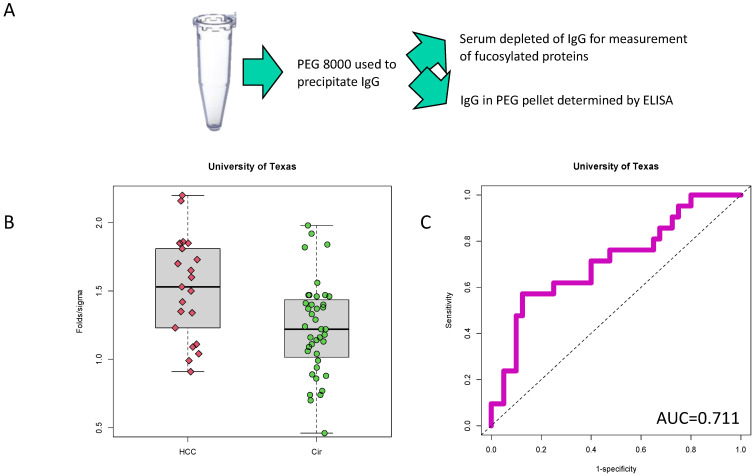
PEG-precipitated IgG is higher in HCC samples with cirrhosis when compared to cirrhosis without HCC. (**A**) Example workflow. PEG-8000 was used to precipitate heterophilic antibodies. The supernatant contained fucosylated glycoproteins of interest while the pellet contained IgG and other immunoglobulins. (**B**) Box and whiskers plot of PEG-IgG in 21 HCC or 40 cirrhosis samples (Cir). (**C**) AUROC of PEG-IgG which has a value of 0.711.

**Figure 2 cancers-14-05970-f002:**
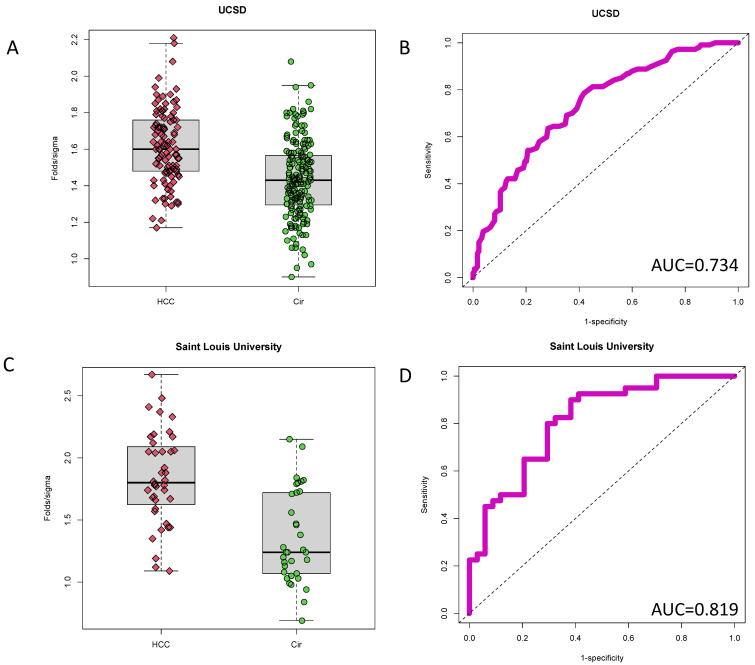
Conformation of PEG-IgG in two other cross-sectional cohorts. (**A**) Box and whiskers plot of PEG-IgG in 107 HCC or 184 cirrhosis samples. (**B**) AUROC of PEG-IgG from cohort shown in panel (**A**), which has a value of 0.734. (**C**) Box and whiskers plot of PEG-IgG in 40 early-stage HCC or 34 cirrhosis samples. (**D**) AUROC of PEG-IgG from cohort shown in panel (**A**), which has a value of 0.819.

**Figure 3 cancers-14-05970-f003:**
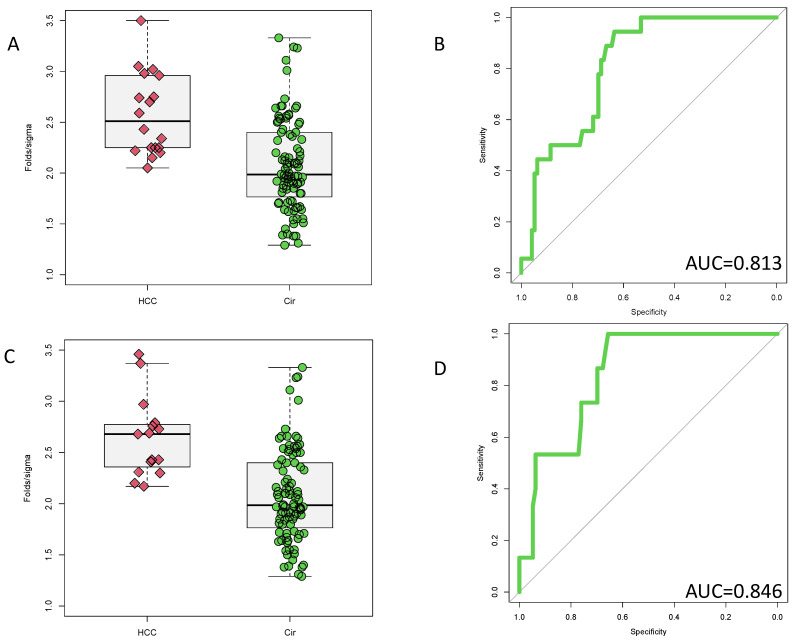
Validation of PEG-IgG in samples collected prospectively but analyzed blinded in a retrospective manner. (**A**) Box and whiskers plot of PEG-IgG in either cirrhotic patients who did not develop HCC or in patients who developed HCC 1–6 months later. (**B**) AUROC of PEG-IgG from cohort shown in panel (**A**), which has a value of 0.813. (**C**) Box and whiskers plot of PEG-IgG in either cirrhotic patients who did not develop HCC or in patients who developed HCC 7+ months later. (**D**) AUROC of PEG-IgG from cohort shown in panel (**A**), which has a value of 0.846.

**Figure 4 cancers-14-05970-f004:**
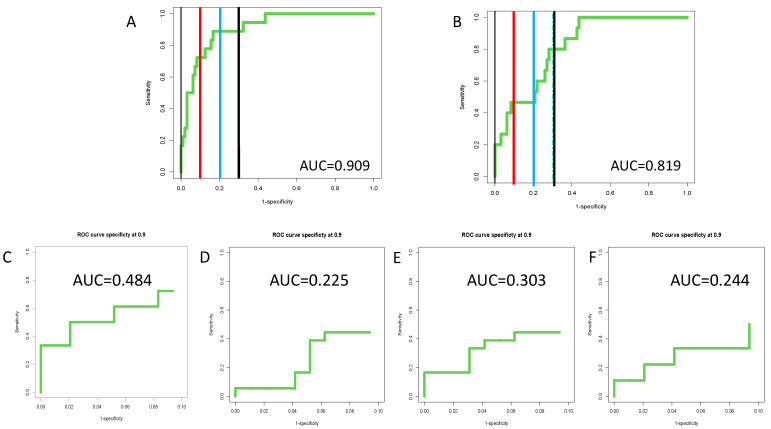
(**A**) Diagnostic algorithm dramatically improves detection of HCC. A diagnostic algorithm was developed using the combined UT, SLU, and UCSD cohorts (training set, *n* = 426). Panels (**A**,**B**) show external validation of the diagnostic algorithm in patients either at 1–6 prior to HCC detection or 7+ months prior to HCC detection. In Panels (**A**,**B**), red lines mark 90% specificity, blue lines mark 80% specificity and black lines mark 70% specificity. Panels (**C**–**F**) show the 90-ROC of the diagnostic algorithm (**C**), PEG-IgG (**D**), AFP (**E**) or fucosylated kininogen (**F**) in the 1–6-month cohort.

**Figure 5 cancers-14-05970-f005:**
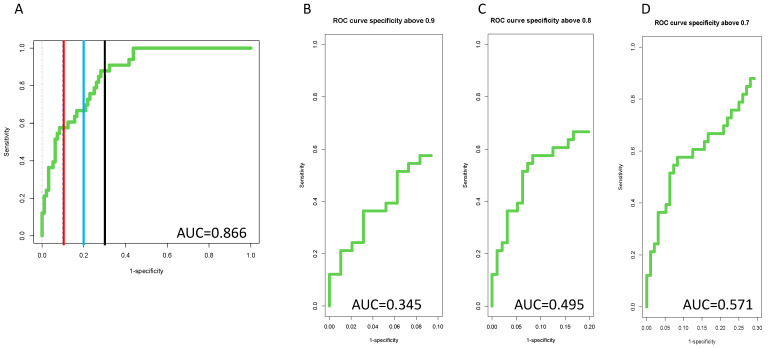
Diagnostic algorithm improves detection of early-stage HCC. (**A**) External validation of the diagnostic algorithm in all patients with early-stage HCC at any time prior to HCC detection. Panels (**B**–**D**) show the 90-ROC, 80-ROC and 70-ROC. In Panel A, the red line mark 90% specificity, the blue line mark 80% specificity and the black line mark 70% specificity.

**Table 1 cancers-14-05970-t001:** Patient Demographics.

	University of Texas ^1^	Saint Louis University ^1^	UCSD ^1^	University of Michigan ^2^
Sample size	61	74	339	97
UCC:Cirrhosis: Other ^3^	21:40:0	40:34:0	107:184:48	37:60
Early stage HCC ^4^	0%	100%	42%	75%
Gender (Male:Female)	38:23	39:35	195:144	60:37
Age (Mean, SD) ^5^	58.23, 6.64	55.86, 11.93	60.32, 10.45	54.46, 8.39
ALK (Mean, SD) ^6^	144.51, 92.75	NA	145.33, 90.88	163.23, 98.57
ALT (Mean, SD) ^7^	NA	NA	47.89, 61.35	59.09, 43.74
AST (Mean, SD) ^8^	69.13, 50.23	NA	66.92, 67.67	79.58, 57.22
AFP (Mean, SD) ^9^	3101.29, 12910.51	65.07, 98.76	169.97, 998.85	13.24, 30.30
PER-IgG (Mean, SD) ^10^	1.32, 0.37	1.62, 0.44	1.50, 0.23	2.24, 0.47

^1^ Three cross sectional sample sets were used in this study. The first, from the University of Texas (UT), consisted of 21 HCC patients with background of cirrhosis and 40 patients with cirrhosis. The second cohort, from St. Louis University (SLU), consisted of 40 HCC patients with background of cirrhosis and 34 patients with cirrhosis. Cohort three, from the University of California at San Diego (UCSD) consisted of 107 patients with HCC in a background of cirrhosis, 184 patients with cirrhosis and 48 patients treated for their HCC, either by liver transplantation, ablation or surgical resection. ^2^ The fourth cohort from the University of Michigan was collected using a prospective specimen-collection, retrospective-blinded-evaluation (PRoBE) design and HCC samples were at a time of 1-6 months prior to HCC or 7+ month prior to HCC. ^3^ Hepatocellular carcinoma or cirrhosis. In all cases, HCC was detected and diagnosed by imaging (CT-Scan or MRI) and cirrhosis was biopsy confirmed. Other includes patients treated for their HCC. ^4^ Early-stage HCC is defined as either UNOS stage 1 or 2 for SLU and UCSD cohorts or Milan Criteria (UT or UM cohorts). ^5^ Mean age in years. ^6^ ALK = alkaline phosphatase. ^7^ ALT = alanine aminotransferase. ^8^ AST = aspartate aminotransferase. ^9^ Alpha-fetoprotein level in ng/mL. ^10^ Relative IgG in PEG-precipitated material as compared to normal human serum.

## Data Availability

The data presented in this study are available on request from the corresponding author.

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
