# Peer review of "A Biomarker Panel Based upon AFP, Fucosylated Kininogen and PEG-Precipitated IgG Is Highly Accurate for the Early Detection Hepatocellular Carcinoma in Patients with Cirrhosis in Phase II and Phase III Biomarker Evaluation"

_cancers, 2022, doi:10.3390/cancers14235970_

Round 1
Reviewer 1 Report
The authors developed an analytical method based on PEG precipitation, which has been applied in biomarker discovery for hepatocellular carcinoma. The work is of great importance. However, I don’t think the authors design the research appropriately. The data and results presented in this study can’t adequately support the conclusion. In addition, there are many English writing mistakes, which has to be improved.
There are many examples:
1. Page 4, line 135 and 136, please use the symbol “°C” for degree rather than a capital C.
2. Page 4, line 158, please add a space in front of “95%”.
3. Page5, line 170, 3-fold rather than 3-folds.
4. Page 5, line 170, to guarantee rather than to guaranteed.
5. Page 5, line 188, delete a.
6. Page 5, line 190, please rephrase “In this assay, following PEG treatment, precipitated IgG was detected using an antibody to IgG via a direct ELISA approach.”
7. Page 5, line 192, delete as.
8. Page 5, line 215, please rephrase “the total IgG detection was higher in patients with HCC as compared to healthy or cirrhotic serum, but only when samples were PEG precipitated.”
Author Response
We apologize for the format of the paper as reviewed. When the paper is submitted, we are unable to check to PDF conversion. This resulted in figures and text being either lost or placed in inappropriate location. We have worked with the editors to address this situation.
In addition, we have addressed many of the grammatical and spelling errors in the manuscript and believe that it has been improved.
1 Page 4, line 135 and 136, please use the symbol “°C” for degree rather than a capital C.
We have corrected this error.
- Page 4, line 158, please add a space in front of “95%”.
We have corrected this error.
- Page5, line 170, 3-fold rather than 3-folds.
We have corrected this error.
- Page 5, line 170, to guarantee rather than to guaranteed.
We have corrected this error.
- Page 5, line 188, delete a.
We have corrected this error.
- Page 5, line 190, please rephrase “In this assay, following PEG treatment, precipitated IgG was detected using an antibody to IgG via a direct ELISA approach.”
We have corrected this error.
- Page 5, line 192, delete as.
We have corrected this error.
- Page 5, line 215, please rephrase “the total IgG detection was higher in patients with HCC as compared to healthy or cirrhotic serum, but only when samples were PEG precipitated.
Reviewer 2 Report
I encourage the authors to make their discussion more robust. Although the fact that PEG-Ig when used as part of the existing algorithm can be used for early detection of HCC is clear, the limitations of the algorithm should be adequately described. The authors should provide more information about the other available biomarkers in HCC, maybe include a section on other approaches that have been suggested to improve HCC surveillance. Shedding light on other novel biomarkers and comparing to their algorithm may be useful
Author Response
Q1: The authors should provide more information about the other available biomarkers in HCC, maybe include a section on other approaches that have been suggested to improve HCC surveillance.
We now discuss several of the other biomarkers and screening modalities that could be used for the early detection of HCC, including several of novel algorithms that are currently being used.
Q2 Shedding light on other novel biomarkers and comparing to their algorithm may be useful.
We thank the reviewer for their kind words and have modified the discussion to include other available biomarkers in HCC as well as a discussion on other approaches that have been suggested to improve HCC surveillance.
Reviewer 3 Report
Overview and general recommendation:
In the manuscript, the authors show higher level of PEG-precipitated IgG in serum of patients with HCC developed from cirrhosis than that of patients with cirrhosis only. This assay can be performed with small amount of serum by ELISA approach. And the authors claim that when combining with other biomarkers such as AFP and fucosylated kininogen, PEG-precipitated IgG will lead to an improved discriminatory ability and algorithm performance.
I find the paper is organized in a proper way and most of the results are well described. The authors perform background research carefully. And major methods are well described in the manuscript and properly used in the research.
I suggest the authors to add some discussion about the significance and novelty of this research. I also suggest the authors to emphasize the importance and the prospect of this research. For example, why this research is more distinguished than other researches. How this research will contribute to diagnose HCC.
Major comments:
1. Figure5 is missing in the manuscript.
Author Response
Q1: I suggest the authors to add some discussion about the significance and novelty of this research.
We thank the reviewer for the kind works and the helpful suggestions. We have modified the discussion to describe both the novelty and significance of the finding but also the potential use of the partial ROCs for biomarker analysis.
Major comments:
- Figure5 is missing in the manuscript.
We apologize for this. When the files are uploaded, we are not given the opportunity to “check the proof” and this error resulted. We are working with the editorial team to ensure and check proper file conversion.
Reviewer 4 Report
In the manuscript submitted by Wang et al, the authors found that the levels of PEG-8000 precipitated IgG were higher in the serum of patients with HCC than that from cirrhotic patients, thus could serve as a new biomarker. Importantly, when the PEG precipitated IgG biomarker was incorporated into an algorithm that consisted of AFP and fucosylated kininogen, it significantly improved the discriminatory ability compared to either maker alone. This conclusion was made after the analysis of three training cohorts and one validation cohort, which consist of 571 patients in total. These results are interesting and significant, which guarantees its publication.
Some specific comments as below:
1. Although the result is significant, the unintelligibility of the working mechanism of PEG precipitation may require high controllability on the sample processing. For example, the authors used 30 min incubation of PEG and serum. Did the authors tested longer or shorter time to optimize the assay?
2. Similarity, based on Figure S2 A, B and D, the increasing percentages of PEG showed increasing beneficial effect. Did the authors test higher percentage of PEG? If not, please explain.
3. In the assays including PEG precipitation and ELISA, did the authors include triplicate or duplicate during the assay?
4. Since the PEG could co-precipitate IgG and IgM, could the authors comment on the level of PEG precipitated IgM?
5. Could the authors comment on the level of fucosylation and/or other glycosylation of the PEG precipitated IgG relative to the overall IgG among the samples?
6. The legend of Figure 5 is missing
7. In the legend of Table 1, the information of 9) and 10) is missing.
Some other minor issues:
8. Please correct the symbol degree (°C but not C)
9. Line 222: ‘We used number of different…’ should be ‘ We used a number of different…’
10. Line 225-226, Is the (AUC=0.616, p=0.1409) direct ELISA result or sandwich ELISA result?
11. Line 384 : there is an extra ‘of’
12. Line 369: Grammar error, please revise.
Author Response
Q1: Although the result is significant, the unintelligibility of the working mechanism of PEG precipitation may require high controllability on the sample processing. For example, the authors used 30 min incubation of PEG and serum. Did the authors tested longer or shorter time to optimize the assay?
We have previously optimized the PEG method for the removal of heterophilic antibodies, without having a negative impact upon the fucosylated glycoproteins of interest.
Q2: Similarity, based on Figure S2 A, B and D, the increasing percentages of PEG showed increasing beneficial effect. Did the authors test higher percentage of PEG? If not, please explain.
We apologize for the confusion regarding this aspect of the assay. As with the time considerations, we previously optimized the PEG method for the removal of heterophilic antibodies, without having a negative impact upon the fucosylated glycoproteins of interest.
Q3: In the assays including PEG precipitation and ELISA, did the authors include triplicate or duplicate during the assay?
We apologize for the omission of this information. Samples were run in duplicate with the average value used for further analysis.
Q4: Since the PEG could co-precipitate IgG and IgM, could the authors comment on the level of PEG precipitated IgM?
This is a very good question and IgM, while it is precipitated,
Q5: Could the authors comment on the level of fucosylation and/or other glycosylation of the PEG precipitated IgG relative to the overall IgG among the samples?
The fucosylated of the PEG precipitated IgG was not altered in HCC patients as compared to patients with cirrhosis, just the total level. We apologize for the confusion on this matter. In contrast, other proteins, such as kininogen were hyper fucosylated in patients with HCC.
Q6: The legend of Figure 5 is missing
We apologize for this. When the files are uploaded, we are not given the opportunity to “check the proof” and this error resulted. We are working with the editorial team to ensure and check proper file conversion.
Q7: In the legend of Table 1, the information of 9) and 10) is missing.
We apologize for this. This has now been corrected.
Q8 Please correct the symbol degree (°C but not C)
This has been corrected.
Q9 Line 222: ‘We used number of different…’ should be ‘ We used a number of different…’
This has been corrected.
Q10 Line 225-226, Is the (AUC=0.616, p=0.1409) direct ELISA result or sandwich ELISA result?
We apologize for the wording and the confusion on this statement. We have modified it to make it clearer.
Q11: Line 384 : there is an extra ‘of’
This has been corrected.
Q12 Line 369: Grammar error, please revise.
This has been corrected.